# Bone Marrow Hypocellularity in Patients with End-Stage Kidney Disease

**DOI:** 10.3390/healthcare9111452

**Published:** 2021-10-27

**Authors:** Chia-Chen Hsieh, Ming-Jen Chan, Yi-Jiun Su, Jen-Fen Fu, I-Kuan Wang, Chao-Yu Chen, Cheng-Hao Weng, Wen-Hung Huang, Ching-Wei Hsu, Tzung-Hai Yen

**Affiliations:** 1Clinical Poison Center, Department of Nephrology, Chang Gung Memorial Hospital, Linkou 333, Taiwan; jack20331@icloud.com (C.-C.H.); b9202066@cgmh.org.tw (M.-J.C.); claire_chaoyu@hotmail.com (C.-Y.C.); drweng@seed.net.tw (C.-H.W.); williammedia@yahoo.com.tw (W.-H.H.); wei2838@gmail.com (C.-W.H.); 2College of Medicine, Chang Gung University, Taoyuan 333, Taiwan; cgfujf@cgmh.org.tw; 3Division of Hematology-Oncology, Department of Internal Medicine, Chang Gung Memorial Hospital, Linkou 333, Taiwan; b9305038@cgmh.org.tw; 4Department of Medical Research, Chang Gung Memorial Hospital, Linkou 333, Taiwan; 5Department of Nephrology, China Medical University Hospital, Taichung 404, Taiwan; ikwang@mail.cmuh.org.tw; 6College of Medicine, China Medical University, Taichung 406, Taiwan

**Keywords:** bone marrow, biopsy, hypocellularity, chronic kidney disease, end-stage kidney disease

## Abstract

Background. Anemia and pancytopenia are not uncommon in patients with chronic kidney disease (CKD). Nevertheless, there is insufficient literature analyzing bone marrow pathology in patients with CKD or end-stage kidney disease (ESKD) receiving dialysis. Methods. This observational cohort study included 22 patients with ESKD and 23 patients with CKD that received bone marrow biopsy and aspiration at Chang Gung Memorial Hospital. Demographic, hematological, and biochemical data were collected at the time of bone marrow study for analysis. Results. Bone marrow aspiration demonstrated that patients with ESKD had a lower percentage of blasts than patients with CKD (0.52 ± 0.84 versus 1.06 ± 0.78 %, *p* = 0.033). Bone marrow biopsy revealed that the overall incidence of hypocellular bone marrow was 55.6%. Furthermore, patients with ESKD had higher proportion of hypocellular bone marrow than patients with CKD (72.7% versus 39.1%, *p* = 0.023). In a multivariate logistic regression model, it was revealed that ESKD status (odds ratio 9.43, 95% confidence interval 1.66–53.63, *p* = 0.011) and megakaryocyte count within bone marrow (odds ratio 0.48, 95% confidence interval 0.29–0.79, *p* = 0.004) were significant predictors for bone marrow hypocellularity. Conclusion. Bone marrow hypocellularity is common in patients with kidney dysfunction. Hypocellular marrow occurs more frequently in patients with ESKD than patients with CKD.

## 1. Introduction

Anemia and pancytopenia are not uncommon in chronic dialysis patients. Nevertheless, there is insufficient literature analyzing bone marrow pathology in patients with chronic kidney disease (CKD) or end-stage kidney disease (ESKD) receiving chronic dialysis (Table 1) [1,2,3,4,5,6]. As early as 1950, Callen et al. [6] studied 102 patients with nephritis, related diseases and hypertension, and found that 80% of patients with CKD suffered hypercellular bone marrow. On the other hand, Eschbach et al. [5] found hypocellular bone marrow in 53.3% of patients with CKD. However, Ho-Yen et al. [4] studied bone marrow biopsies of 15 patients with CKD compared with those of a control group matched for age and sex. Although the percentage of two groups was not mentioned, there was no significant difference in bone marrow cellularity. In patients with ESKD receiving chronic dialysis, Sikole et al. [3] reported that 60% of patients suffered hypocellular bone marrow. Finally, Weng et al. [2] also reported that hypocellular bone marrow was present in 51.3% of ESRD patients undergoing chronic dialysis.

Taiwan is an epidemic area of kidney disease. The motivation for this research was due to a significant, but as yet unsatisfactorily answered question that occurred in many CKD or ESKD patients receiving chronic dialysis at our hospital. Some of our patients suffered peripheral blood cytopenia, but other patients did not have cytopenia. Theoretically, blood cells are produced in the bone marrow. Thus, this brings up an important question of what the bone marrow pathology of the uremic patients is.

Hypocellular bone marrow with pancytopenia could be caused by aplastic anemia, inherited bone marrow failure syndromes, drugs, toxins, infections, nutritional disorders, or autoimmune disorders [7]. Therefore, the objective of this study was to analyze bone marrow changes in patients with CKD and/or ESKD receiving dialysis.

## 2. Materials and Methods

### 2.1. Patients

Between 2001 and 2020, a total of 22 patients with ESKD undergoing chronic dialysis (aged 17.0–80.0 years, 59.1% male) and 23 patients with CKD (aged 39.0–81.0 years, 39.1% male) received a bone marrow biopsy and aspiration at the Chang Gung Memorial Hospital, Linkou, Taiwan. Baseline demographic, hematological, and biochemical data including hemogram, kidney function, liver function, ferrokinetics, lipid and electrolyte profiles were obtained at the time of the bone marrow study for cross-sectional analysis.

### 2.2. Inclusion and Exclusion Citeria 

All patients with ESKD or CKD who underwent bone marrow biopsy were recruited for this study. Patients who had diagnoses of hematological malignancies that affected the blood, bone marrow, and lymph nodes such as leukemia, multiple myeloma, amyloidosis, and myelodysplastic syndrome were excluded from analysis.

### 2.3. Definitions and ESKD and CKD

CKD was defined by the presence of kidney damage or estimated glomerular filtration rate (eGFR) of less than 60 mL/min per 1.73 m^2^ for more than 3 months [8]. CKD Stage 3 included patients with an eGFR of 30–59 mL/min/1.73 m^2^, and CKD Stage 4 patients have an eGFR of 15–29 mL/min/1.73 m^2^. CKD Stage 5 included patients with an eGFR of <15 mL/min/1.73 m^2^. ESKD was the last stage of CKD, and the term was used to indicate those patients who undergo chronic dialysis treatment.

### 2.4. Bone Marrow Aspiration and Biopsy Procedure

The hemostatic profile of the patients was checked before examination. The procedure was performed under local analgesia with lateral decubitus position. The usual site for bone marrow examination was the posterior iliac crest if the patient was mobile without bony lesion. Methods for the preparation, processing, and reporting of bone marrow aspirates and biopsy specimens were executed according to published guidelines [9].

### 2.5. Definition of Bone Marrow Cellularity

Bone marrow pathology report was the gold standard to evaluate bone marrow cellularity [10]. Hematopoietic stem cells and stromal cells (generally adipocytes) were contained in bone marrow, and marrow cellularity was defined as the volume ratio of hematopoietic cells and adipocytes. The range of normal cellularity in adult hematopoietic bone marrow was 30 to 70% that varied according to patient’s age. Hypercellular marrow was defined as cellularity of more than 70%, normocellular marrow as ratio 30 to 70%, and hypocellular marrow as under 30% bone marrow [9].

### 2.6. Laboratory

Laboratory analysis, including hemogram, kidney function, liver function, ferrokinetics, lipid and electrolyte profiles were measured by automated and standardized methods. The complete blood count with differential was quantified by automated cell counter, blood urea nitrogen, alanine aminotransferase and total cholesterol by enzymatic method, creatinine, albumin, triglyceride, iron and ferritin by colorimetric method, calcium, high and low density lipoprotein by spectrophotometric method, phosphate, sodium, and potassium by ion-selective electrode method, total iron binding capacity by nephelometry method, glycated hemoglobin by high performance liquid chromatography.

### 2.7. Statistical Analysis

Data were presented as mean ± standard deviation or number and percentage in parentheses. Student’s t-test was used for parametric variables and Chi-Square or Fisher’s exact test, for non-parametric variables. Univariate binary logistic regression analysis was conducted to analyze the potential risk factors for bone marrow hypocellularity. To control for confounding factors, multivariate binary logistic regression analysis was completed to analyze the significant risk factors on univariate analysis. A *p* value of less than 0.05 was designated as the significance level to reject the null hypothesis. Analyses were completed using IBM SPSS Statistics Version 20.0 (IBM Corp., New York, NY, USA).

## 3. Results

The indications for bone marrow examination were presented in Table 2. Unexplained anemia (88.9%) was the most common indication for bone marrow biopsy in both the ESKD (81.8%) and CKD (95.7%). There were also no significant differences in the indications for a biopsy between the ESKD and CKD (*p* = 0.146).

The mean age of patients underwent bone marrow examination was 57.0 ± 16.4 years with roughly equal sex distribution (48.9% male, Table 3). A large proportion of patients (91.1%) had hypertension. Diabetes mellitus (46.7%) was the major cause of renal dysfunction in the patients. Most patients (73.3%) were in Stage 5 of CKD. Furthermore, patients with ESKD were younger compared to patients with CKD (52.1 ± 18.6 versus 61.7 ± 12.6 years, *p* = 0.047). 

As shown in Table 4, patients with ESKD had higher blood urea nitrogen (83.4 ± 30.2 versus 63.1 ± 29.5, *p* = 0.041), creatinine (11.4 ± 3.1 versus 4.1 ± 2.0 mg/dL, *p* < 0.001), calcium level (9.4 ± 1.3 versus 8.2 ± 0.8 mg/dL, *p* = 0.008), and lower estimated glomerular filtration rate (4.7 ± 1.5 versus 16.9 ± 12.9 cc/min, *p* < 0.001) compared to patients with CKD. Furthermore, patients with ESKD had lower hemoglobin (7.3 ± 1.3 versus 8.4 ± 1.6 g/dL, *p* = 0.017), hematocrit (21.9 ± 3.9 versus 25.0 ± 4.9 %, *p* = 0.025), and red blood cell count (2.4 ± 0.4 versus 2.8 ± 0.6 10^6^/μL, *p* = 0.046) than patients with CKD.

The reports of bone marrow aspiration are presented in Table 5. Patients with ESKD had lower percentage of blasts than patients with CKD (0.5 ± 0.8 versus 1.1 ± 0.8 %, *p* = 0.033).

The reports of bone marrow biopsies are presented in Table 6. The percentage of hypocellular bone marrow was 55.6% (Table 6). Patients with ESKD had higher percentage of hypocellular bone marrow than patients with CKD (72.7% versus 39.1%, *p* = 0.023).

In a multivariate logistic regression model (Table 7), it was shown that ESKD status (odds ratio 9.43, 95% confidence interval 1.66–53.63, *p* = 0.011) and megakaryocyte counts within bone marrow (odds ratio 0.48, 95% confidence interval 0.29–0.79, *p* = 0.004) were significant predictors for bone marrow hypocellularity.

## 4. Discussion

Although abnormal hematological profiles are prevalent in patients with kidney dysfunction, limited literature has investigated bone marrow pathology in uremic populations. This is supposed to be the first study analyzing bone marrow pathology of patients with CKD and ESKD at the same time. Although bone marrow biopsy is crucial for identifying causes for peripheral blood cytopenia in patients with kidney dysfunction, the procedure is not routinely performed in clinical practice due to its invasive nature and potential risk [11]. Only a few studies with small sample sizes explored bone marrow pathology, and the results also varied widely (Table 1). Therefore, this analytical data is important as it confirmed that bone marrow hypocellularity was common (55.6%) in patients with kidney dysfunction. Furthermore, hypocellular marrow occurred more frequently in patients with ESKD compared with patients with CKD (72.7% versus 39.1%, *p* = 0.023).

There is no clear-cut explanation for bone marrow hypocellularity in patients with kidney dysfunction. Accumulation of uremic toxin may have deleterious effects on bone marrow cells [12]. Potential inhibition of erythropoiesis by uremic sera has been reported by an in vitro study [13]. Indoxyl sulfate, one of soluble uremic toxins accumulated during CKD, is known to impair erythropoiesis via a hypoxia-induced factor pathway [14]. In addition, indoxyl sulfate also induces eryptosis via extracellular calcium entry [15]. Indoxyl sulfate also cause erythrocyte death via organic anion transporter 2, nicotinamide adenine dinucleotide phosphate oxidase, and glutathione-independent mechanisms [16]. Such toxicity not only affects the mature erythrocyte, but also affects progenitor cells. Other uremic toxins may also interfere erythropoiesis. Polyamines have been demonstrated to reduce proliferation and maturation of erythroid precursor cells [17]. Acrolein, one of the oxidation products of polyamines, may also promote suicidal erythrocyte death via cell membrane scrambling and cell shrinkage [18]. Burst-forming unit-erythroid proliferation rate is reduced by uremic serum. However, such inhibition of proliferation attenuates if the uremia patient is treated with hemodiafiltration with endogenous reinfusion, a method known to have better removal of uremic toxin [19]. Nevertheless, toxic effects of uremic toxin on bone marrow are still poorly understood. 

In this study, patients with ESKD suffered lower hemoglobin (*p* = 0.017), hematocrit (*p* = 0.025) and red blood cell count (*p* = 0.046) than patients with CKD. Anemia in patients with ESKD receiving chronic dialysis could be explained by chronic blood loss, decreased erythrocyte production, or increased erythrocyte destruction. The kidneys are the chief organs of erythropoietin production in adults, while in fetuses the liver is the primary organ of erythropoietin gene expression [20]. With advancing renal impairment, erythropoietin production decreases progressively and serves as the foremost reason for anemia in uremic patients. Another possibility is the shortening lifespan of red blood cells after erythropoietin therapy. In a study, Sato el al [21] found that the life span of red blood cells was lesser in hemodialysis patients than healthy controls (89 ± 28 vs. 128 ± 28 days), and life span was negatively correlated with erythropoiesis-stimulating agent (ESA) doses. Other factors such as iron deficiency, blood loss during hemodialysis, inflammation, and shortages of nutrition could also lead to anemia [22]. In this study, all patients with ESKD received ESA but none of the patients with CKD received ESA. In addition, iron profile and bone marrow iron store were not different between ESKD and CKD patients. Therefore, the greater degree of anemia in ESKD than CKD may be explained by impaired response to ESA because bone marrow hypocellularity was greater in ESKD than CKD. Further study is required to validate this finding.

The bone marrow biopsy report showed that megakaryocyte count was 3.4 ± 2.1 per high power field (Table 6). Furthermore, megakaryocyte count within bone marrow was a significant predictor for bone marrow hypocellularity (*p* = 0.004, Table 7). In other words, there was a 0.48 reduced risk of developing hypocellular marrow for each increase in 1 megakaryocyte per high power field. In the bone marrow, platelets are produced by megakaryocytes, which are regulated by thrombopoietin [23]. Although thrombocytopenia was described in patients with ESKD, the megakaryocyte counts in the bone marrow aspirate were not reduced [24]. The bone marrow aspiration report also showed that most of the patients had normal (71.1%) or increased (6.7%) megakaryocyte distribution (Table 5). Similarly, the number of megakaryocytes was normal or increased in the bone marrow of patients with immune thrombocytopenia [25]. The mechanisms of platelet underproduction in immune thrombocytopenia remain uncertain. Likewise, there is no clear explanation for the association between increased megakaryocyte count within bone marrow and hypocellular marrow. Further studies are warranted. 

Diabetes mellitus, chronic glomerulonephritis, and hypertension accounted for 46.7%, 28.9%, and 22.2% of causes of renal insufficiency in this study (Table 2). The causes of CKD varied between different countries, ethnicities, and age. Overall, diabetes mellitus was the leading cause of CKD, which was 44% in United States, 27.5% in United Kingdom [26]. In contrast, the greatest cause of CKD in China was chronic glomerulonephritis, which was 57.4% of patients [27]. In Taiwan, the epidemiology distribution of CKD was similar to that in the United States, which was 43.2% with diabetes mellitus, 25.1% with chronic glomerulonephritis, and 8.3% with primary hypertension [28].

Patients with ESKD had higher calcium levels (*p* = 0.008) compared to patients with CKD. The normal kidney regulates serum calcium and phosphate level via control of intestinal absorption and renal tubular excretion. Once the regulation of the intestine and kidney are interrupted by renal impairment, the balance of serum calcium, phosphate level, parathyroid hormone, vitamin D metabolism may be disturbed. The biochemical findings of mineral bone disease included increment of fibroblast growth factor, parathyroid hormone elevation, and serum phosphate and decrement of 1,25-dihydroxyvitamin D and serum calcium [29,30,31]. With advancing kidney impairment, active vitamin D deficiency worsens and contributes to hypocalcemia and secondary hyperparathyroidism. Osteoclast activity is stimulated and finally results in abnormalities of bone turnover. Therefore, phosphate-lowering therapy and maintenance of serum calcium level are mandatory for patients of ESKD. In Taiwan, calcium-based phosphate binders are routinely used in ESKD patients because the cost is payable by National Health Insurance. Another consideration is the calcium-sensing receptor (CaSR). Several links exist between hematopoietic pathways and mineral bone metabolism, and CaSR plays an important role in the homing of hematopoietic stem cells [32]. In CaSR knockout mice, bone hypocellularity causing a reduction in hematopoietic stem cell number was observed. Defects in the adherence of CaSR deficient hematopoietic stem cells to bone marrow extracellular matrix were reported [33]. In the setting of uremia, decreased expression of CaSR was not only observed in parathyroid gland cells, but also in blood cells [34,35]. Nevertheless, the complex associations between hypercalcemia, CaSR, and bone marrow hypocellularity needs further research.

The major limitation of this retrospective observational cohort study is its small sample size. Furthermore, none of the patients received renal biopsy for confirmation of their causes of renal dysfunction. The control of anemia was not good in either group, so other factors may have influenced hemoglobin control in these patients. Nevertheless, this study is limited by lack of intact parathyroid hormone, urine protein creatinine ratio, alfa2 globulins and total protein determination. Another limitation is the lack of protocol for bone marrow biopsies and aspirations. All patients in this study received a one-off bone marrow study. None of the patients received repeated bone marrow studies for illustrating sequential pathological changes within the bone marrow. However, considering the invasiveness and complications of bone marrow procedures, protocol bone marrow biopsy and aspiration would be difficult. 

## 5. Conclusions

With the paucity of medical literature, this study is important to elucidate bone marrow changes in patients with renal dysfunction. The analytical results show that bone marrow hypocellularity is common in patients with kidney dysfunction. Hypocellular marrow occurs more frequently in patients with ESKD than patients with CKD. Furthermore, ESKD status and megakaryocyte count within bone marrow are significant predictors of bone marrow hypocellularity.

## Figures and Tables

**Table 1 healthcare-09-01452-t001:** Published literature on bone marrow pathology in patients with kidney dysfunction.

Study	Year	Area	Sample Size	Patient Population	Hypocellular Marrow (%)
Current study	2021	Taiwan	45	ESKD, CKD	72.7 (ESKD), 39.1 (CKD)
Latif et al. [1]	2017	Pakistan	57	CKD	1.75
Weng et al. [2]	2015	Taiwan	78	ESKD	51.3
Sikole et al. [3]	1997	North Macedonia	32	ESKD	60
Ho-Yen et al. [4]	1980	Scotland	30	CKD	No hypocellular marrow
Eschbach et al. [5]	1970	United States	90	CKD	53.3
Callen et al. [6]	1950	United States	102	CKD	80% hypercelluarity

Note: CKD chronic kidney disease, ESKD end-stage kidney disease.

**Table 2 healthcare-09-01452-t002:** Indications for bone marrow examination, stratified by renal function as ESKD or CKD (*n* = 45).

Indications for Bone Marrow Examination	All Patients (*n* = 45)	Patients with ESKD (*n* = 22)	Patients with CKD (*n* = 23)	*p* Value
				0.146
Unexplained anemia, *n* (%)	40 (88.9)	18 (81.8)	22 (95.7)	
Pancytopenia, *n* (%)	3 (6.7)	3 (13.6)	0 (0)	
Thrombocytopenia, *n* (%)	1 (2.2)	1 (2.2)	0 (0)	
Fever of unknown origin, *n* (%)	1 (2.2)	0 (0)	1 (4.3)	

**Table 3 healthcare-09-01452-t003:** Baseline characteristics of patients, stratified by renal function as ESKD or CKD (*n* = 45).

Variable	All Patients (*n* = 45)	Patients with ESKD (*n* = 22)	Patients with CKD (*n* = 23)	*p* Value
Demographics				
Age (year) [range]	57.0 ± 16.4 [17.0–81.0]	52.1 ± 18.6 [17.0–80.0]	61.7 ± 12.6 [39.0–81.0]	0.047 *
Male, *n* (%)	22 (48.9)	13 (59.1)	9 (39.1)	0.181
Body mass index (kg/m^2^)	23.4 ± 4.9	23.6 ± 3.7	23.2 ± 5.8	0.796
Hypertension, *n* (%)	41 (91.1)	21 (95.5)	20 (87.0)	0.317
Coronary artery disease, *n* (%)	7 (15.6)	3 (13.6)	4 (17.4)	0.728
Alcohol consumption, *n* (%)	2 (4.4)	1 (4.5)	1 (4.3)	0.974
Smoking habit, *n* (%)	2 (4.4)	1 (4.5)	1 (4.3)	0.974
Betel nut chewing, *n* (%)	1 (2.2)	1 (4.5)	0 (0)	0.301
Causes of kidney dysfunction, *n* (%)				0.686
Diabetes mellitus	21 (46.7)	9 (40.9)	12 (52.2)	
Hypertension	10 (22.2)	5 (22.7)	5 (21.7)	
Glomerulonephritis	13 (28.9)	7 (31.8)	6 (26.1)	
Unknown	1 (2.2)	1 (4.5)	0 (0)	
Stage of CKD, *n* (%)				<0.001 *
Stage 3	2 (4.4)	0 (0)	2 (8.7)	
Stage 4	10 (22.2)	0 (0)	10 (43.5)	
Stage 5	33 (73.3)	22 (100)	11 (47.8)	

Note: * *p* < 0.05.

**Table 4 healthcare-09-01452-t004:** Laboratory findings of patients, stratified by renal function as ESKD or CKD (*n* = 45).

Variable	All Patients (*n* = 45)	Patients with ESKD (*n* = 22)	Patients with CKD (*n* = 23)	*p* Value
Estimated glomerular filtration rate (cc/min)	11.0 ± 11.0	4.7 ± 1.5	16.9 ± 12.9	<0.001 ***
Blood urea nitrogen (mg/dL)	74.0 ± 31.2	83.4 ± 30.2	63.1 ± 29.5	0.041 *
Creatinine (mg/dL)	7.7 ± 4.5	11.4 ± 3.1	4.1 ± 2.0	<0.001 ***
Alanine aminotransferase (U/L)	29.0 ± 29.2	33.5 ± 31.0	23.4 ± 26.6	0.296
Calcium (mg/dL)	9.0 ± 1.3	9.4 ± 1.3	8.2 ± 0.8	0.008 *
Phosphorus (mEq/L)	5.5 ± 1.6	5.8 ± 1.5	5.0 ± 1.7	0.169
Sodium (mEq/L)	137.4 ± 4.0	136.4 ± 4.4	139.5 ± 1.8	0.043 *
Potassium (mEq/L)	4.4 ± 0.7	4.2 ± 0.6	4.6 ± 0.6	0.054
Albumin (g/dL)	3.6 ± 0.4	3.7 ± 0.5	3.5 ± 0.3	0.290
Glycated hemoglobin (%)	5.8 ± 1.0	5.7 ± 0.7	6.0 ± 1.4	0.429
High density lipoprotein (mg/dL)	41.6± 12.2	39.5 ± 11.3	44.8 ± 13.3	0.251
Low density lipoprotein (mg/dL)	119.9 ± 90.0	120.1 ± 89.4	119.7 ± 94.2	0.992
Total cholesterol (mg/dL)	176.81 ± 73.22	173.45 ± 73.39	181.0 ± 75.20	0.763
Triglyceride (mg/dL)	169.6 ± 275.4	137.7 ± 111.3	212.1 ± 404.9	0.437
Iron (μg/dL)	84.3 ± 54.0	86.8 ± 59.2	78.1 ± 41.6	0.709
Total iron binding capacity (μg/dL)	259.1 ± 58.0	267.0 ± 55.0	241.3 ± 64.3	0.306
Iron/Total iron binding capacity (μg/dL)	0.3 ± 0.2	0.3 ± 0.2	0.4 ± 0.2	0.807
Ferritin (ng/mL)	996.5 ± 963.3	1038.9 ± 881.0	919.4 ± 1139.8	0.747
White blood cells (10^3^/uL)	5.8 ± 2.7	5.8 ± 3.1	5.8 ± 2.4	0.965
Red blood cell (10^6^/uL)	2.6 ± 0.6	2.5 ± 0.4	2.8 ± 0.6	0.046 *
Hemoglobin (g/dL)	7.9 ± 1.6	7.3 ± 1.3	8.4 ± 1.6	0.017 *
Hematocrit (%)	23.4 ± 4.6	21.9 ± 3.9	25.0 ± 4.9	0.025 *
Mean corpuscular volume (fL)	90.1 ± 9.8	89.1 ± 10.2	91.1 ± 9.6	0.517
Mean corpuscular hemoglobin (pg/cell)	30.1 ± 3.6	29.9 ± 3.8	30.3 ± 3.4	0.765
Mean corpuscular hemoglobin concentration (g/dL)	33.6 ± 1.2	33.6 ± 1.2	33.6 ± 1.3	0.983
Red blood cell distribution width (%)	15.7 ± 3.7	16.6 ± 4.0	14.8 ± 3.1	0.121
Platelet (10^3^/uL)	157.5 ± 87.2	153.9 ± 89.0	161.1 ± 87.3	0.787
Neutrophil (%)	67.9 ± 12.7	66.4 ± 13.5	69.2 ± 12.1	0.519
Lymphocyte (%)	19.5 ± 8.3	17.6 ± 7.9	21.2 ± 8.4	0.198
Monocyte (%)	7.4 ± 4.3	8.2 ± 4.5	6.7 ± 4.1	0.283
Eosinophil (%)	3.1 ± 3.3	2.7 ± 2.1	3.4 ± 4.2	0.540
Basophil (%)	0.4 ± 0.6	0.3 ± 0.4	0.5 ± 0.7	0.313

Note: * *p* < 0.05, *** *p* < 0.001.

**Table 5 healthcare-09-01452-t005:** Reports of bone marrow aspiration (*n* = 45).

Variable	All Patients (*n* = 45)	Patients with ESKD (*n* = 22)	Patients with CKD (*n* = 23)	*p* Value
Cellularity				0.011 *
Hypercellularity or normocellularity, *n* (%)	23 (51.1)	7 (31.8)	16 (69.6)	
Hypocellularity, *n* (%)	22 (48.9)	15 (68.2)	7 (30.4)	
Megakaryocyte distribution				0.107
Absent, *n* (%)	1 (2.2)	1 (4.5)	0 (0)	
Decreased, *n* (%)	9 (20.0)	7 (31.8)	2 (8.7)	
Normal, *n* (%)	32 (71.1)	12 (54.5)	20 (87.0)	
Increase, *n* (%)	3 (6.7)	2 (9.1)	1 (4.3)	
Morphology, megakaryocytes				0.201
Normal, *n* (%)	43 (95.6)	20 (90.9)	23 (100)	
Dysplasia, *n* (%)	1 (2.2)	1 (4.5)	0 (0)	
Absent, *n* (%)	1 (2.2)	1 (4.5)	0 (0)	
Myeloid/erythroid ratio	2.1 ± 1.4	2.3 ± 1.8	1.8 ± 0.8	0.285
Myeloid series, *n* (%)	49.6 ± 14.3	50.7 ± 15.2	48.6 ± 13.7	0.627
Blast, *n* (%)	0.8 ± 0.9	0.5 ± 0.8	1.1 ± 0.8	0.033 *
Promyelocyte, *n* (%)	1.7 ± 1.8	1.8 ± 2.0	1.6 ± 1.7	0.717
Myelocyte and metamyelocyte, *n* (%)	16.4 ± 6.6	17.7 ± 7.4	15.3 ± 5.5	0.226
Band and segmental cell, *n* (%)	31.2 ± 12.3	30.8 ± 12.2	30.7 ± 11.9	0.980
Morphology of myeloid series				0.139
Normal	43 (95.6)	20 (90.9)	23 (100)	
Abnormal	2 (4.4)	2 (9.1)	0 (0)	
Erythroid series (%)	30.4 ± 13.3	32.0 ± 16.6	28.8 ± 9.3	0.424
Morphology of erythroid series				0.139
Normal	43 (95.6)	20 (90.9)	23 (100)	
Abnormal	2 (4.4)	2 (9.1)	0 (0)	
Monohistiocyte (%)	0.9 ± 0.9	1.1 ± 1.2	0.8 ± 0.7	0.307
Eosinophil (%)	2.8 ± 1.8	2.4 ± 1.7	3.2 ± 1.8	0.132
Plasma cell (%)	1.8 ± 2.4	1.3 ± 1.6	2.3 ± 2.8	0.151
Lymphoid cell (%)	13.0 ± 6.9	13.0 ± 8.1	13.0 ± 5.5	0.969
Sideroblast, grade, *n* (%)				0.462
Normal	5 (11.4)	2 (9.5)	3 (13.0)	
Increase	19 (43.2)	7 (33.3)	12 (52.2)	
Decrease	6 (13.6)	3 (14.3)	3 (13.0)	
Absent	14 (31.8)	9 (42.9)	5 (21.7)	
Iron staining, grade, *n* (%)				0.086
Grade 0	7 (15.9)	2 (9.5)	5 (21.7)	
Grade 1	2 (4.5)	1 (4.8)	1 (4.3)	
Grade 2	3 (6.8)	0 (0)	3 (13.0)	
Grade 3	19 (43.2)	14 (66.7)	5 (21.7)	
Grade 4	10 (22.7)	3 (14.3)	7 (30.4)	
Grade 5	2 (4.5)	1 (4.8)	1 (4.3)	
Grade 6	1 (2.3)	0 (0)	1 (4.3)	

Note: * *p* < 0.05.

**Table 6 healthcare-09-01452-t006:** Reports of bone marrow biopsy (*n* = 45).

Variable	All Patients (*n* = 45)	Patients with ESKD (*n* = 22)	Patients with CKD (*n* = 23)	*p* Value
Cellularity				0.023 *
Hypercellularity or normocellularity, *n* (%)	20 (44.4)	6 (27.3)	14 (60.9)	
Hypocellularity, *n* (%)	25 (55.6)	16 (72.7)	9 (39.1)	
Myeloid/erythroid ratio	2.2 ± 1.2	2.3 ± 1.5	2.1 ± 1.0	0.626
Megakaryocyte, per high power field	3.4 ± 2.1	3.5 ± 2.0	3.4 ± 2.2	0.867
Reticulin staining, grade				
Grade 0	42 (93.3)	20 (90.9)	22 (95.7)	0.524
Grade 1	3 (6.7)	2 (9.1)	1 (4.3)	
Iron staining, grade, *n* (%)				0.297
Grade 0	8 (17.8)	4 (18.2)	4 (17.4)	
Grade 1	3 (6.7)	2 (9.1)	1 (4.3)	
Grade 2	24 (53.3)	9 (40.9)	15 (65.2)	
Grade 3	7 (15.6)	5 (22.7)	2 (8.7)	
Grade 4	2 (4.4)	2 (9.1)	0 (0)	
Grade 5	1 (2.2)	0 (0)	1 (4.3)	
Grade 6	0 (0)	0 (0)	0 (0)	

Note: * *p* < 0.05.

**Table 7 healthcare-09-01452-t007:** Analysis of predictors for bone marrow hypocellularity (*n* = 45).

Variable	Univariate Logistic Regression Analysis			Multivariate Logistic Regression Analysis		
	Odds Ratio	95% Confidence Interval	*p* Value	Odds Ratio	95% Confidence Interval	*p* Value
ESKD status (yes)	4.149	1.179–14.493	0.027 *	9.43	1.66–53.63	0.011 *
Age (per 1 year increase)	0.98	0.94–1.02	0.233			
Sex (male)	1.32	0.41–4.31	0.641			
BMI (per 1 kg/m^2^ increase)	0.91	0.79–1.06	0.222			
Coronary artery disease (yes)	0.93	0.18–4.72	0.927			
Hypertension (yes)	0.78	0.10–6.10	0.815			
Alcohol consumption (yes)	1.26	0.07–21.74	0.872			
Betel nut chewing (yes)	1.26	0.07–21.74	0.872			
Estimated glomerular filtration rate (per 1 cc/min increase)	0.98	0.93–1.04	0.589			
Blood urea nitrogen (per 1 mg/dL increase)	1.02	0.99–1.04	0.147			
Creatinine (per 1 mg/dL increase)	1.14	0.98–1.32	0.085			
Alanine aminotransferase (per 1 U/L increase)	1.03	0.99–1.07	0.138			
Calcium (per 1 mg/dL increase)	1.12	0.64–1.96	0.701			
Phosphorus (per 1 mEq/L increase)	1.33	0.82–2.18	0.251			
Sodium (per 1 mEq/L increase)	1.06	0.88–1.26	0.533			
Potassium (per 1 mEq/L increase)	1.02	0.38–2.75	0.966			
Albumin (per 1 g/dL increase)	3.08	0.52–18.18	0.217			
Glycated hemoglobin (per 1% increase)	0.73	0.30–1.74	0.472			
High density lipoprotein (per 1 mg/dL increase)	1.00	0.94–1.06	0.886			
Low density lipoprotein (per 1 mg/dL increase)	0.995	0.986–1.004	0.265			
Total cholesterol (per 1 mg/dL increase)	0.99	0.98–1.00	0.147			
Triglyceride (per 1 mg/dL increase)	1.00	0.991–1.004	0.378			
Iron (per 1 μg/dL increase)	1.01	0.99–1.02	0.354			
Total iron binding capacity (per 1 μg/dL increase)	1.00	0.98–1.01	0.725			
Iron/Total iron binding capacity (per 1 μg/dL increase)	1.00	0.99–1.01	0.548			
Ferritin (per 1 ng/mL increase)	1.001	1.000–1.002	0.138			
White blood cells (per 10^3^/uL increase)	0.79	0.62–1.02	0.067			
Red blood cell (per 1 × 10^6^ /uL increase)	0.29	0.07–1.15	0.078			
Hemoglobin (per 1 g/dL increase)	0.70	0.45–1.09	0.115			
Hematocrit (%) (per 1% increase)	0.91	0.80–1.05	0.189			
Mean corpuscular volume (per 1 fL increase)	1.03	0.97–1.10	0.322			
Mean corpuscular hemoglobin (per 1 pg/cell increase)	1.06	0.89–1.26	0.516			
Mean corpuscular hemoglobin concentration (per 1 g/dL increase)	0.73	0.44–1.25	0.262			
Red blood cell distribution width (per 1% increase)	1.02	0.86–1.20	0.858			
Platelet (per 1 × 10^3^ /uL increase)	0.994	0.987–1.001	0.106			
Neutrophil (per 1% increase)	0.95	0.90–1.01	0.099			
Lymphocyte (per 1% increase)	1.00	0.93–1.09	0.888			
Monocyte (per 1% increase)	0.92	0.92–1.28	0.316			
Eosinophil (per 1% increase)	0.96	0.79–1.18	0.715			
Basophil (per 1% increase)	1.06	0.33–3.37	0.924			
Bone marrow aspiration						
Myeloid/erythroid ratio (per of 1 increase)	1.09	0.70–1.68	0.708			
Myeloid series (per 1% increase)	0.99	0.95–1.04	0.752			
Blast (per 1% increase)	0.86	0.43–1.73	0.571			
Promyelocyte (per 1% increase)	1.07	0.76–1.51	0.685			
Myelocyte and metamyelocyte (per 1% increase)	0.97	0.89–1.07	0.550			
Band and segmental cell (per 1% increase)	1.00	0.95–1.05	0.972			
Morphology of myeloid series (normal)	1.26	0.07–21.74	0.872			
Erythroid series (per 1% increase)	0.99	0.95–1.04	0.745			
Morphology of erythroid series (normal)	1.31	0.07–21.74	0.872			
Monohistiocyte (per 1% increase)	1.82	0.75–4.41	0.183			
Eosinophil (per 1% increase)	1.00	0.71–1.41	0.986			
Plasma cell (per 1% increase)	0.88	0.66–1.17	0.374			
Lymphoid cell (per 1% increase)	1.00	0.91–1.09	0.908			
Iron staining (per 1 grade increase)	1.23	0.82–1.85	0.316			
Bone marrow biopsy						
Myeloid/erythroid ratio (per 1 increase)	0.96	0.59–1.58	0.878			
Megakaryocyte (per 1 cell high power field increase)	0.59	0.40–0.86	0.005 **	0.48	0.29–0.79	0.004 **
Reticulin staining (per 1 grade increase)	2.67	0.22–32.26	0.438			
Iron staining (per 1 grade increase)	1.25	0.84–1.84	0.268			

Note: * *p* < 0.05, ** *p* < 0.01.

## Data Availability

The datasets used and analyzed in this study are available from the corresponding author upon request.

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
