# Peer review of "Bone Marrow Hypocellularity in Patients with End-Stage Kidney Disease"

_healthcare, 2021, doi:10.3390/healthcare9111452_

Round 1

Reviewer 1 Report

Dear,

Comments follow throughout the attached document.

Author Response

Please clarify, with chronic kidney disease and/or end-stage kidney disease?

Response: Thank you for the comment. The sentence has been revised.

Therefore, the goal of this study was to examinebone marrow changes in patients withCKD and/or ESKD receiving dialysis, and to inspect the correlations of bone marrow pathology with clinical and laboratory parameters.

I think the introduction would be more complete, with the pathophysiological characterization of chronic kidney disease, end-stage kidney disease and Bone Marrow Hypocellularity. Furthermore, to define more specifically the associated biochemical and hematological parameters among this type of patients evaluated in this study.

Response: Thank you for the comment. The paragraphs have been revised.

Anemia and pancytopenia are not uncommon in chronic dialysis patients. Nevertheless, there are insufficient literatures analyzing bone marrow pathology in patients with CKD or end-stage kidney disease (ESKD) receiving chronic dialysis (Table 1). [1-6]As early as 1950, Callen et al [6]studied 102 patients with nephritis, related diseases and hypertension, and found that 80% of patients with CKD suffered hypercellular bone marrow. On the other hand, Eschbach et al [5]found hypocellular bone marrow in 53.3% of patients with CKD. However, Ho-Yen et al [4]studied bone marrow biopsies of 15 patients with CKDcompared with those of a control group matched for age and sex. Although the percentage of two groups was not mentioned, there was no significant difference in bone marrow cellularity. In patients with ESKD receiving chronic dialysis, Sikole et al [3]presented that 60% of the patients suffered hypocellular bone marrow. Finally, Weng et al [2]also reported that hypocellular bone marrow presented in 51.3% of ESRD patients undergoing chronic dialysis.

Taiwan is an epidemic area of kidney disease. The motivation for this research was due to a significant, but as yet unsatisfactorily answered question that occurred in many CKD or ESKD patients receiving chronic dialysis at our hospital. Some of our patients suffered peripheral blood cytopenia, but other patients did not have cytopenia. Theoretically, blood cells are produced in the bone marrow. Thus, this brings up an important question of what the bone marrow pathology of the uremic patients is.

Hypocellular bone marrow with pancytopenia could be caused by aplastic anemia, inherited bone marrow failure syndromes, drugs, toxins, infections, nutritional disorders or autoimmune disorders. [7]Therefore, the objective of this study was to analyzebone marrow changes in patients withCKD and/or ESKD receiving dialysis.

This study was submitted to an ethics committee? Study participants were asked an informed consent?

Response: Thank you for the comment. The Institutional Review Board Statement has been provided at the end of manuscript.

Institutional Review Board Statement: This retrospective observational cohort study adhered to the guidelines of the Declaration of Helsinki and was approved by the Medical Ethics Committee of Chang Gung Memorial Hospital. The Institutional Review Board number was 202000663B0.

Informed Consent Statement: Since this study involved a retrospective review of existing data, Institutional Review Board approval was obtained without specific informed consent from the patients. However, informed consent was obtained from all patients before bone marrow biopsy. All individual information was securely protected by delinking identifying information from main data set and was only available to investigators. Furthermore, all of the data were analyzed anonymously. The Institutional Review Board of the Chang Gung Memorial Hospital had specifically waived the need for consent.

Please define the age ranges and gender of the individuals studied.

Response: Thank you for the comment. The age ranges and gender have been defined.

Between 2001 and 2020, a total of 22 patients with ESKD undergoing chronic dialysis (aged 17.0 – 80.0 years, 59.1% male) and 23 patients with CKD (aged 39.0 – 81.0 years, 39.1% male) received bone marrow biopsy and aspiration at Chang Gung Memorial Hospital, Linkou, Taiwan.

What criteria were used to define whether or not patients had CKD and ESKD?

Response: Thank you for the comment. The definitions for CKD and ESKD have been included.

CKD was defined by the presence of kidney damage or estimated glomerular filtration rate (eGFR) of less than 60 ml/min per 1.73 m2 for more than 3 months. [8]CKD Stage 3 includedpatients with an eGFR of 30 - 59 ml/min/1.73 m2, and CKD Stage 4 patients have an eGFR of 15 - 29 ml/min/1.73 m2. CKD Stage 5 included patients with an eGFR of < 15 ml/min/1.73 m2. ESKDwas the last stage of CKD, and the term was used to indicate those patients who undergo chronic dialysis treatment.

Please define which parameters are evaluated, and the procedure used for its determination?

Response: Thank you for the comment. The parameters and the procedure used for analysis have been described.

Baseline demographic, hematological and biochemical data including hemogram, kidney function, liver function, ferrokinetics, lipid and electrolyte profiles were obtained at the time of the bone marrow study for cross-sectional analysis.

Laboratory analysis, including hemogram, kidney function, liver function, ferrokinetics, lipid and electrolyte profiles were measured by automated and standardized methods.

Please put gender.

Response: Thank you for the comment. The gender data has been included.

The mean age of patients underwent bone marrow examination was 57.0 ± 16.4 years with roughly equal sex distribution (48.9% male).

The determination and procedures for these parameters must be defined in the methodology.

Response: Thank you for the comment.

Laboratory analysis, including hemogram, kidney function, liver function, ferrokinetics, lipid and electrolyte profiles were measured by automated and standardized methods.

In the previous paragraph you refer: "Accumulation of uremic toxin may have deleterious effect on bone marrow cell." However here justify the hematopoietic alterations in renal patients in the terminal phase, with problems in erythropoietin production. Please clarify these statements.

Response: Thank you for the comment. The statements have been clarified.

Anemia in patients with ESKD receiving chronic dialysis could be explained by chronic blood loss, decreased erythrocyte production, or increased erythrocyte destruction. The kidneys are the chief organs of erythropoietinproduction in adults, while in the fetuses the liver is the primary organ of erythropoietingene expression. [20]With advancing renal impairment, the erythropoietin production decreased progressively and served as the foremost reason for anemia in uremic patients. Another possibility was the shortening lifespan of red blood cells after erythropoietin therapy. In a study, Sato el al [21]found that the life spans of red blood cells was lesser in hemodialysis patients than healthy controls (89 ± 28 versus 128 ± 28 days, and the life span was negatively correlated with erythropoiesis-stimulating agent (ESA) doses. Other factors such as iron deficiency, blood loss during hemodialysis, inflammation, and shortage of nutrition could also lead to anemia. [22]In this study, all patients from ESKD received ESA but none of the patients from CKD received ESA. In addition, iron profile and bone marrow iron store were not different between ESKD and CKD patients. Therefore, the greater degree of anemia in ESKD than CKD may be explained by impaired response to ESA because bone marrow hypocellularity was greater in ESKD than CKD. Further study is required to validate this finding.

I do not understand this relationship, it is stated that in this study "Megakaryocytes count within bone marrow was a significant predictor for bone marrow hypocellularity (P = 0.004)." and here in this study (21) it is stated that the megakaryocyte counts in ESKD patients are not reduced, please clarify.

Response: Thank you for the comment. The paragraph has been revised.

The bone marrow biopsy report showed that megakaryocyte count was 3.4 ± 2.1 per high power field(Table 6).Furthermore, megakaryocytes count within bone marrow was a significant predictor for bone marrow hypocellularity (P = 0.004, Table 7). In other words, there was a 0.48 reduced risk of developing hypocellular marrow for each increase in 1 megakaryocyte per high power field. In the bone marrow, platelets are produced by megakaryocyte, which is regulated by thrombopoietin. [23]Although thrombocytopenia was described in patients with ESKD, the megakaryocyte counts in the bone marrow aspirate were not reduced. [24]The bone marrow aspiration report also showed that most of the patients were having normal (71.1%) or increased (6.7%) megakaryocyte distribution (Table 5). Similarly, the number of megakaryocytes is normal or increased in the bone marrow of patients with immune thrombocytopenia. [25]The mechanisms of platelet underproduction in immune thrombocytopenia remain uncertain. Likewise, there is no clear explanation for the association between increased megakaryocyte count within bone marrow and hypocellular marrow. Further studies are warranted.

The results and their discussion are poor and very repetitive and do not respond to the proposed objectives, namely the indicated correlations. Do not discuss the relationship between biochemical parameters and renal dysfunction, why?

Response: Thank you for the comment. The Results and Discussion paragraphs have been thoroughly revised. The mechanisms of hypocellular marrow, anemia, megakaryocyte count within bone marrow, causes of renal dysfunction, and biochemical disturbance such as calcium are discussed. Please advise us again if we miss out anything.

Would it be expected?

Response: Thank you for the comment. Yes, as expected. Anemia and pancytopenia are not uncommon in patients with CKDand ESKD undergoing chronic dialysis.

In what sense, the low number of megakaryocytes

Response: Thank you for the comment.

The bone marrow biopsy report showed that megakaryocyte count was 3.4 ± 2.1 per high power field(Table 6).Furthermore, megakaryocytes count within bone marrow was a significant predictor for bone marrow hypocellularity (P = 0.004, Table 7). In other words, there was a 0.48 reduced risk of developing hypocellular marrow for each increase in 1 megakaryocyte per high power field. In the bone marrow, platelets are produced by megakaryocyte, which is regulated by thrombopoietin. [23]Although thrombocytopenia was described in patients with ESKD, the megakaryocyte counts in the bone marrow aspirate were not reduced. [24]The bone marrow aspiration report also showed that most of the patients were having normal (71.1%) or increased (6.7%) megakaryocyte distribution (Table 5). Similarly, the number of megakaryocytes is normal or increased in the bone marrow of patients with immune thrombocytopenia. [25]The mechanisms of platelet underproduction in immune thrombocytopenia remain uncertain. Likewise, there is no clear explanation for the association between increased megakaryocyte count within bone marrow and hypocellular marrow. Further studies are warranted.

Reviewer 2 Report

The Authors report the results of bone marrow analysis in patients with renal disease.

This field of research is very interesting since literature on this topic is lacking.

There are some critical points to be highlighted:
1. The case series of the study is poor: only 45 patients, an important element which is underlined in the discussion.

2. The reason for bone marrow biopsy is not reported. Since BM biopsy is an invasive examination, it should be done only if there is a clinical reason and not only for research. Moreover, whether the study was submitted to an ethical committee is not reported and is essential.

3. The causes of renal disease are not reported in detail. Patients may have different causal nephropathies, which may induce different states of inflammation (vasculitis?, lupus?, interstitial disease?, etc.).

4. The classification of CKD stages of the patients examined is not reported. The mean of serum creatinine is 4.1 mg/dl, so I suppose that most patients must have been in advanced stages, what makes a lot of difference in anemia control for example.

5. The Authors state in the conclusions that all patients with ESKD received erythropoietin and none in the CKD one. However the control of anemia is not good in either groups, so other factors may have influenced the Hb control in these patients.

The patients look like not very well clinically controlled, what is also suggested by the high levels of calcium in ESKD (which suggests a third hyperparathyroidism and not only Calcium-based phosphate binders). Moreover, levels of ferritin are also high, which leads to think that an inflammatory/malnutrition condition was present. This suggests again that the clinical condition of the patients was not well corrected. So, other data must be added: PTH, P, PCR, alfa2 globulins, total proteins, albumin.

6. Minor point is “hyper” instead of “hypocellularity” while citing Callen work.

Author Response

The Authors report the results of bone marrow analysis in patients with renal disease.

This field of research is very interesting since literature on this topic is lacking.

Response: Thank you for the comment.

There are some critical points to be highlighted:
1. The case series of the study is poor: only 45 patients, an important element which is underlined in the discussion.

Response: Thank you for the comment. This limitation has been stressed in the Discussion paragraph.

The major limitation of this retrospective observational cohort study is small sample size. Furthermore, none of the patients received renal biopsy for confirmation of causes of renal dysfunction.The control of anemia is not good in either group, so other factors may have influenced the hemoglobin control in these patients. Nevertheless, this study is limited by lack of intact parathyroid hormone, urine protein creatinine ratio, alfa2 globulins and total proteins determination. Another limitation is lacking protocol bone marrow biopsy and aspiration. All patients in this study received one time of bone marrow study. None of the patients received repeated bone marrow study for illustrating sequential pathological changes within bone marrow. However, considering the invasiveness and complications of bone marrow procedure, the protocol bone marrow biopsy and aspiration would be difficult. 

  1. The reason for bone marrow biopsy is not reported. Since BM biopsy is an invasive examination, it should be done only if there is a clinical reason and not only for research. Moreover, whether the study was submitted to an ethical committee is not reported and is essential.

Response: Thank you for the comments. The indications for bone marrow examination were presented in Table 2. Unexplained anemia (88.9 %) was the most common indication for bone marrow biopsy in both the ESKD (81.8 %) and CKD (95.7 %). There were also no significant differences in the indications for a biopsy between the ESKD and CKD (P = 0.146).

The Institutional Review Board Statement has been provided at the end of manuscript.

Institutional Review Board Statement: This retrospective observational cohort study adhered to the guidelines of the Declaration of Helsinki and was approved by the Medical Ethics Committee of Chang Gung Memorial Hospital. The Institutional Review Board number was 202000663B0.

Informed Consent Statement: Since this study involved a retrospective review of existing data, Institutional Review Board approval was obtained without specific informed consent from the patients. However, informed consent was obtained from all patients before bone marrow biopsy. All individual information was securely protected by delinking identifying information from main data set and was only available to investigators. Furthermore, all of the data were analyzed anonymously. The Institutional Review Board of the Chang Gung Memorial Hospital had specifically waived the need for consent.

  1. The causes of renal disease are not reported in detail. Patients may have different causal nephropathies, which may induce different states of inflammation (vasculitis?, lupus?, interstitial disease?, etc.).

Response: Thank you for the comment. We understand that the causes of kidney dysfunction are inadequate. Clinically, none of the patients had autoimmune disease such as vasculitis or collagen vascular disease such as systemic lupus erythematous. None of the patients received renal biopsy for confirmation of causes of renal dysfunction. The limitation has been included in the Discussion paragraph.

The major limitation of this retrospective observational cohort study is small sample size. Furthermore, none of the patients received renal biopsy for confirmation of causes of renal dysfunction.The control of anemia is not good in either group, so other factors may have influenced the hemoglobin control in these patients. Nevertheless, this study is limited by lack of intact parathyroid hormone, urine protein creatinine ratio, alfa2 globulins and total proteins determination. Another limitation is lacking protocol bone marrow biopsy and aspiration. All patients in this study received one time of bone marrow study. None of the patients received repeated bone marrow study for illustrating sequential pathological changes within bone marrow. However, considering the invasiveness and complications of bone marrow procedure, the protocol bone marrow biopsy and aspiration would be difficult. 

  1. The classification of CKD stages of the patients examined is not reported. The mean of serum creatinine is 4.1 mg/dl, so I suppose that most patients must have been in advanced stages, what makes a lot of difference in anemia control for example.

Response: Thank you for the comment. The classification of CKD stages of the patients has been included (Table 3). Most patients (73.3%) were in Stage 5 of CKD.

  1. The Authors state in the conclusions that all patients with ESKD received erythropoietin and none in the CKD one. However the control of anemia is not good in either group, so other factors may have influenced the Hb control in these patients.

The patients look like not very well clinically controlled, what is also suggested by the high levels of calcium in ESKD (which suggests a third hyperparathyroidism and not only Calcium-based phosphate binders). Moreover, levels of ferritin are also high, which leads to think that an inflammatory/malnutrition condition was present. This suggests again that the clinical condition of the patients was not well corrected. So, other data must be added: PTH, P, PCR, alfa2 globulins, total proteins, albumin.

Response: Thank you for the comment. We understand that the control of anemia is not good in either group (Table 4), so other factors may have influenced the hemoglobin control in these patients. The blood intact parathyroid hormone level of patients with ESKD was 402.8± 395.3 pg/dL. However, the blood test report was not available in patients with CKD. The blood levels of phosphorus and albumin were shown in Table 4. Nevertheless, there were no data for urine protein creatinine ratio, alfa2 globulins or total proteins.These limitation have been included in the Discussion paragraph.

The major limitation of this retrospective observational cohort study is small sample size. Furthermore, none of the patients received renal biopsy for confirmation of causes of renal dysfunction.The control of anemia is not good in either group, so other factors may have influenced the hemoglobin control in these patients. Nevertheless, this study is limited by lack of intact parathyroid hormone, urine protein creatinine ratio, alfa2 globulins and total proteins determination. Another limitation is lacking protocol bone marrow biopsy and aspiration. All patients in this study received one time of bone marrow study. None of the patients received repeated bone marrow study for illustrating sequential pathological changes within bone marrow. However, considering the invasiveness and complications of bone marrow procedure, the protocol bone marrow biopsy and aspiration would be difficult. 

  1. Minor point is “hyper” instead of “hypocellularity” while citing Callen work.

Response: Thank you for reminding us.

As early as 1950, Callen et al [6]studied 102 patients with nephritis, related diseases and hypertension, and found that 80% of patients with CKD suffered hypercellular bone marrow.

Round 2

Reviewer 1 Report

The manuscript has been significantly improved, however it is still necessary to clarify some aspects of the methodology.

Comments follow throughout the attached document.
